# Long-Term Surgical Results of Cortical Mastoid Bone Osteomas

Giulia Donati and Luca Oscar Redaelli de Zinis *

Department of Medical and Surgical Specialties, Radiological Sciences and Public Health, Section of Audiology, University of Brescia, 25100 Brescia, Italy; ju.donati@hotmail.it
* Correspondence: luca.redaellidezinis@unibs.it

**Abstract:** Background: Though osteomas can commonly arise in the cranial bones, an extra canalicular mastoid bone location is a rare entity with less than 200 cases described to date. We present three cases of cortical mastoid bone osteomas and compare them with cases presented in the literature. Methods: In this study, we used a retrospective chart analysis. Results: All three patients presented after years of progressively increasing postauricular swelling without symptoms. Temporal bone non-contrast CT allowed accurate preoperative diagnosis. Surgical treatment was performed for cosmetic issues with minimal postoperative morbidity. Complete excision was achieved in all cases, and to date, there is no evidence of recurrence. Conclusions: Mastoid osteomas are rare benign slow-growing tumors. They usually present as a painless cosmetic disfigurement and are otherwise asymptomatic. Surgical excision is the treatment of choice when they cause esthetic discomfort or are symptomatic. Recurrences are infrequently reported.

**Keywords:** mastoid; extracanalicular; cortical; temporal bone; osteoma

## 1. Introduction

Osteomas are osteoblastic tumors, benign in nature and with a slow growth capacity [1,2]. They are frequently encountered in cranial bones, mostly in the frontal and ethmoidal bones, while they are rare in the temporal bone, with less than 200 cases described in the literature and a reported incidence of 0.1–1% of all benign tumors of the skull [2]. Within the temporal bone, the most frequent localization is the external auditory canal (EAC, external acoustic meatus), while extra canalicular osteomas are rare entities: the mastoid process is a frequent location, other locations are the squama, middle ear structures, promontory, internal auditory canal (IAC, internal acoustic meatus), and styloid process [1–4].

We present three cases of mastoid bone osteomas treated at the ENT Department of the University Hospital of Brescia.

## 2. Materials and Methods

We conducted a retrospective chart review of 3 cases of mastoid bone osteoma treated at the ENT Department of the University Hospital of Brescia between 1992 and 1995.

## 3. Results

### 3.1. Case 1

A 24-year-old woman presented with a history of slowly increasing left-sided postauricular swelling noted a few years before. There was no history of trauma, ear infection, otologic problems, or other symptoms. On examination, the mass was hard, fixed to deep planes, and measured about 3 cm in diameter. It caused no pain. The overlying skin showed no signs of inflammation. Otoscopy and pure tone audiometry were normal. A temporal bone CT scan revealed a bony mass arising from the external cortex of the left mastoid region that was 2.5 cm in diameter (Figure 1). The surrounding soft tissues were separated, and the auricle was anteriorized. There was no inflammatory involvement of the mastoid or IAC abnormalities. Surgical excision was planned for esthetic purposes.

The tumor was exposed by postauricular skin incision and soft tissue dissection. It had a pedicle, was covered by a layer of periosteum, and was firmly attached to the external mastoid cortex. To demarcate the lesion from normal mastoid air cells, its margins were drilled. The tumor was about 3.0 × 2.0 cm, hard, and white in appearance. Soft tissues and skin were closed in layers. Histopathologic examination was suggestive of "fibrous osteoma". Postoperative recovery was uneventful. To date, there are no signs of recurrence.

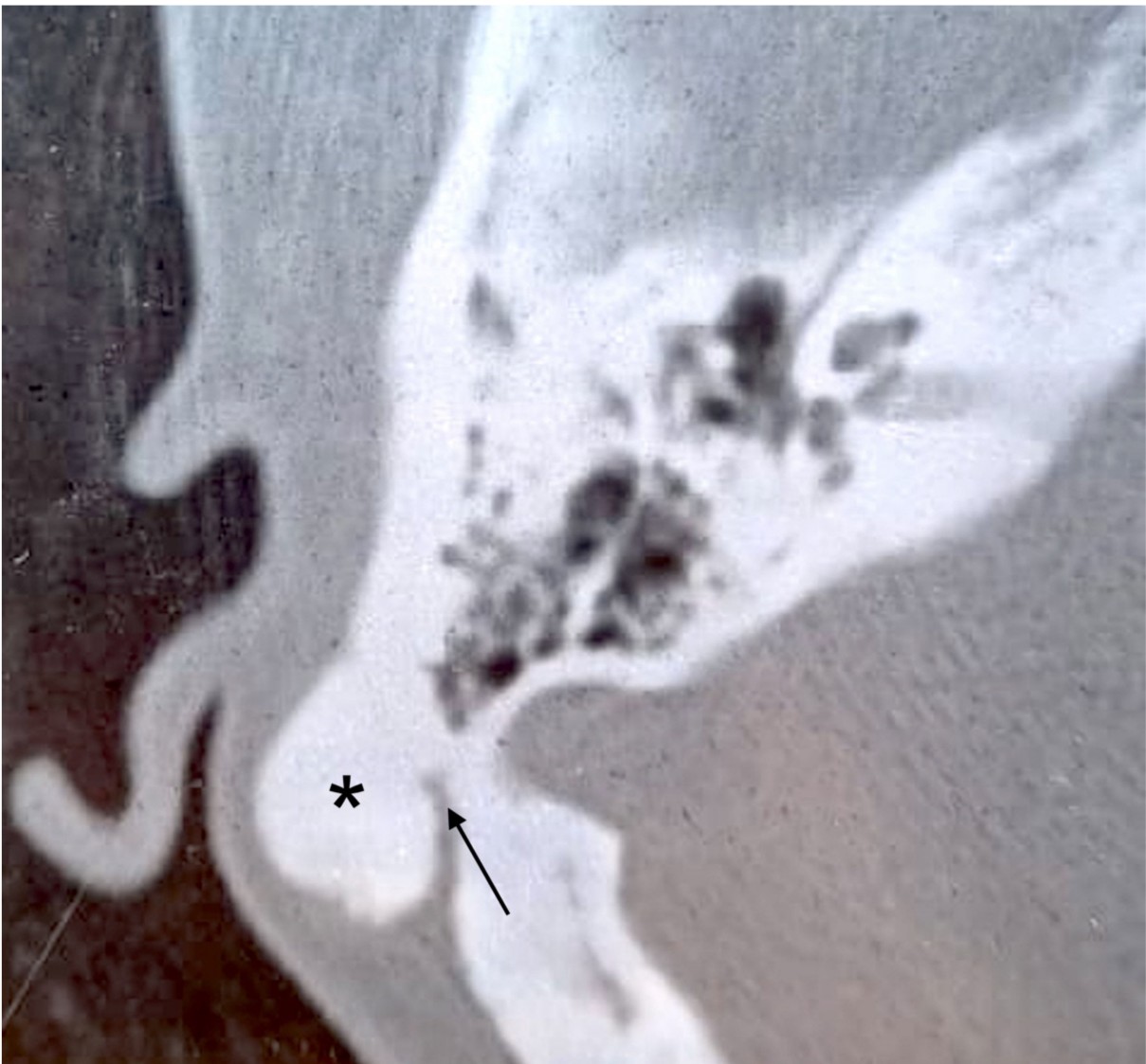

**Figure 1.** Axial CT scan. Pedicled bony mass (asterisk) arising from mastoid cortex of the right side. The pedicle is indicated by the arrow.

*3.2. Case 2*

A 30-year-old woman was first seen in our department because she noticed a slowly increasing right-sided postauricular hard swelling, otherwise asymptomatic, some months before. There was no history of trauma, ear infection, otologic problems, or other symptoms. On examination, the mass measured about 2.5 × 3.0 cm in diameter, was hard, fixed to the deep planes, and covered by normal skin. Micro-otoscopic examination revealed a normal tympanic membrane and EAC, and no anatomical abnormalities. Pure tone audiometry showed a normal threshold in the left ear and mild sensorineural hearing loss in the right ear, limited to medium and high frequencies. The Auditory Brainstem Response Test (ABR) was bilaterally normal. No focal neurological defects were found on neurological

examination. Temporal bone CT was suggestive of a mastoid osteoma, with its apex directed to the inferior third of the mastoid, which was well-pneumatized. The lesion measured 3.0 × 2.0 cm in diameter. The external half of the mass was homogeneously osteosclerotic, while the deepest component showed an irregular pattern within bony trabeculae (Figure 2). The middle ear, EAC, and IAC were normal. The patient underwent surgery for cosmetic issues. A right incision was performed 2 cm behind the postauricular sulcus, extending from the temporal line to the mastoid tip. Subperiosteal dissection of soft tissues and the upper part of the sternocleidomastoid muscle was needed to expose the tumor. Its margins were drilled to separate the mass from the surrounding well-pneumatized mastoid air cells. The lesion was oval-shaped, composed of compact bone, and located lateral to the sigmoid sinus. Depression drainage was placed, and soft tissues and skin were closed in layers. The histopathologic diagnosis was "osteoma", with macroscopic homogeneous nodular features, hard consistency, and thickened cortex, while microscopic examination revealed Haversian canals within the compact bone. The postoperative course was complicated by hemotympanum and consequential conductive hearing loss, which recovered in a few days. To date, there are no signs of recurrence.

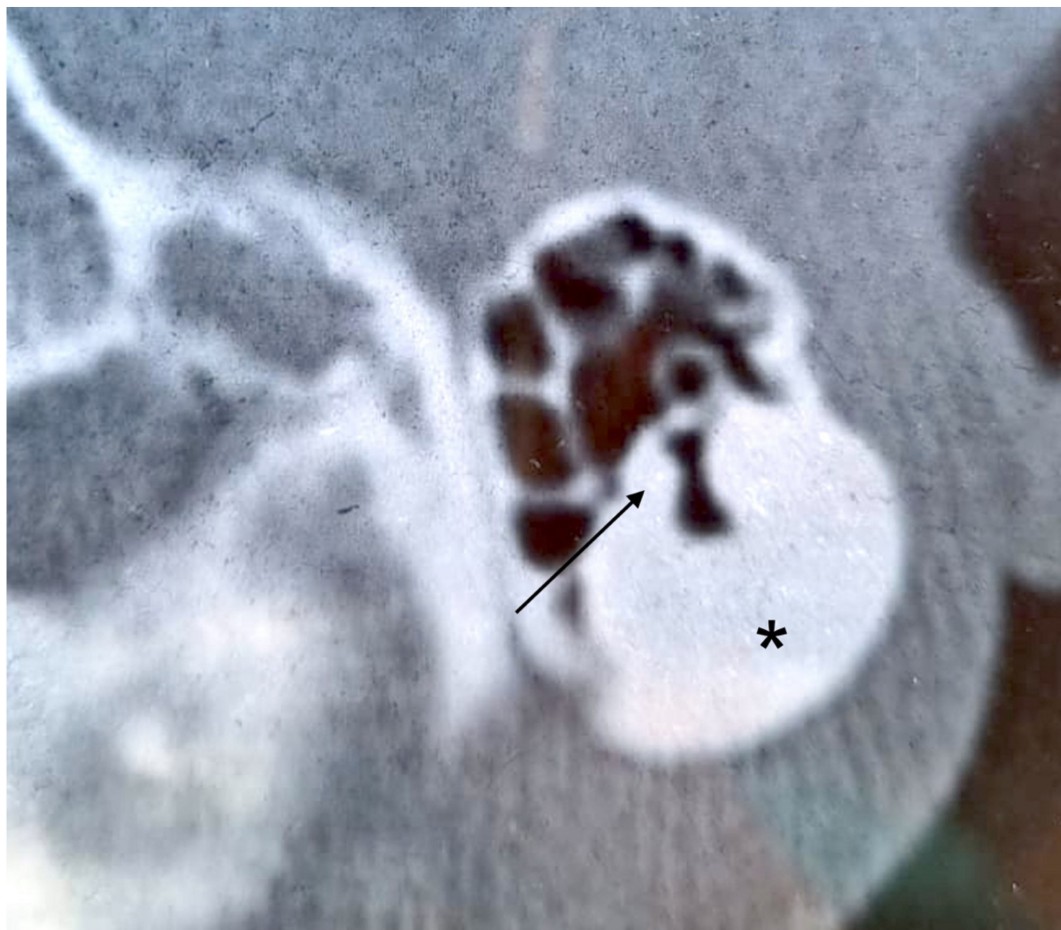

**Figure 2.** Axial CT scan. Left mastoid cortex osteoma (asterisk) with trabeculae in the deepest component (arrow).

### 3.3. Case 3

A 34-year-old man was referred due to a history of progressively increasing right-sided temporal swelling since the age of 8 years. He did not complain of pain or ear discharge and referred no previous trauma or infection. On examination, a hard postauricular mass about 2.5 cm in diameter, fixed to the deep planes and covered by normal skin, was found. Otoscopy and pure tone audiometry were normal. A temporal bone CT scan showed a

right-sided mastoid bony mass that was 2.4 cm in diameter. It appeared to originate from the periosteum and had a homogeneous appearance, with well-defined margins (Figure 3). There were no signs of any infiltration of surrounding soft tissues. The patient underwent surgery for cosmetic reasons. A postauricular incision 5 cm in extension and soft tissue dissection revealed a hard, ivory-colored mass originating from the outer cortex of the mastoid bone. The tumor was completely excised with the help of cutting burrs, diamond burrs, and ear curette. The soft tissues and skin were closed in layers. The histopathologic diagnosis was "osteoma". Postoperative recovery was uneventful. To date, there are no signs of recurrence.

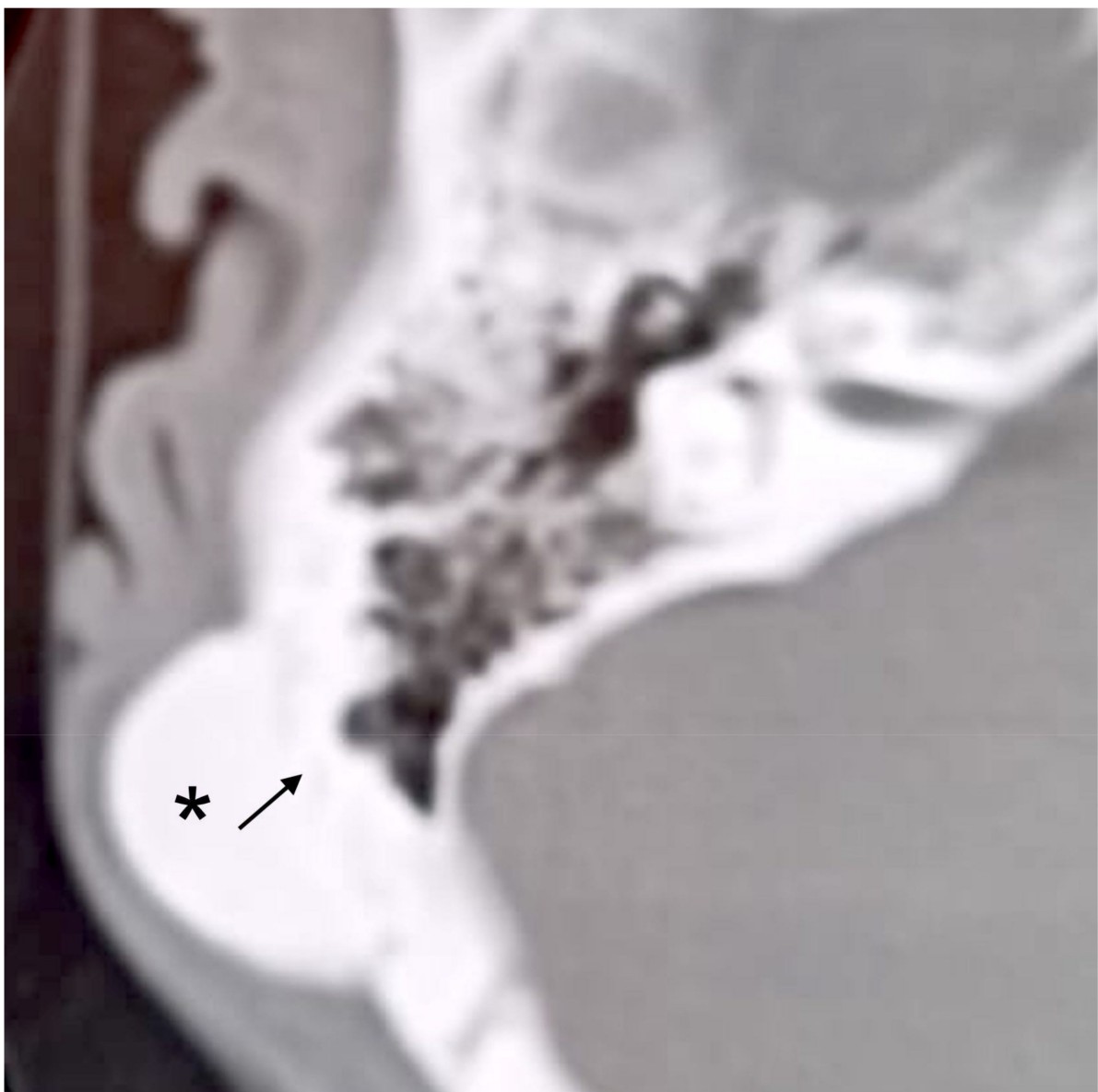

**Figure 3.** Axial CT scan. Right mastoid cortex osteoma (asterisk) with a subtle line margin (arrow).

### 4. Discussion

#### 4.1. Presentation

Osteomas are slow-growing, benign, bony tumors that are histologically composed of mature lamellar bone, with Haversian-like canals, with a predominance in young females [2,4]. Typically, they are solitary lesions growing from the outer surface of the cortex, giving rise to external swelling [5]. However, they can present as multiple osteomas as in the mastoid cavity case, growing from the internal table, as reported by Marrocco

in 1948 [6]. Osteomas usually involve the craniofacial skeleton being the most frequent benign tumor of the skull and facial bones. Typically, they affect the frontal and ethmoid sinuses, and rarely the EAC, eustachian tube, petrous apex, glenoid fossa, styloid process, middle ear, and ossicular chain [1,7–9]. Mastoid osteomas represent 0.1% to 1% of all benign tumors of the cranial bone and the mastoid represents the most frequent localization within temporal bone [2,9]. In fact, according to Denia, of 53 cases of temporal bone osteomas reported in the literature up to 1979, 41 were mastoid osteomas (78%), followed by the squama and the IAC, whereas other sites are extremely rare [9]. Osteomas grow very slowly with long-time stability. They have a smooth surface, with hard consistency, and do not involve the skin [5]. Mastoid osteomas are frequently asymptomatic, except for the slow-growing esthetic deformity, but may cause symptoms of intracranial compression or extension, as in the case reported by Lima Guarneri in 2019 [10]. Earache or neck tenderness can be reported for contiguity to the greater auricular and small occipital nerves [9]. The tumor may also dislocate the posterior meatal wall, leading to conductive hearing loss, or can occasionally extend to the internal table of the temporal bone, causing tenderness on pressure [5,9]. Osteomas can rarely be part of a syndrome such as Gardner syndrome, where multiple osteomas are associated with, subcutaneous fibromas and lipomas, desmoid tumors, epidermal cysts, and intestinal polyposis that carries a high risk of malignant transformation [5,7]. According to a review by Domínguez Pérez et al. [11], from 1861 to 2011, a total of 137 cases of mastoid osteomas have been described, to which about an additional 25 cases in the last 10 years should be included. With our case series, we can add three more cases.

### 4.2. Origin

The etiology of temporal bone osteomas has not been established. According to some authors, they develop from preosseous connective tissue [2,12], while others suggest trauma as a potential inciting factor, with subsequent ossifying periostitis, in addition to surgery, radiotherapy, chronic inflammation, pituitary gland dysfunction, and congenital mechanisms [2,13–15]. However, we cannot define the cause of the cases described herein.

### 4.3. Histopathological Classification

Osteomas are classified into out- and ingrowing types and unilateral and bilateral types. Pathology of osteomas includes the following four types [4,6]:

- Compact or eburneum (the most common type): this type develops from the mastoid cortex but can involve mastoid air cells, and histologically, it is composed of dense, lamellated bone tissue, with few vessels;
- Cancellous (rare): this lesion consists of fibrous cellular tissue and cancellous bone;
- Cartilaginous (uncommon): this type consists of bone and cartilage;
- Mixed (uncommon): this tumor consists of a mixture of types of bone found in compact and cancellous osteomas.

The presented cases were two compact osteomas (cases 2 and 3) and one cancellous osteoma (case 1).

### 4.4. Diagnosis and Differential Diagnosis

As mastoid osteomas are rare lesions, frequently asymptomatic, and with a certain histopathological variability, accurate preoperative radiological evaluation is of utmost importance for differential diagnosis and early identification of potential complications. The imaging technique for the diagnosis of bone lesions of the mastoid process is CT [3,7,16–18]. On CT mastoid osteomas appear as well-demarcated outgrowing lesions from the external cortex. Compact osteomas are sessile, uniform lesions similar to normal cortical bone. Cancellous osteomas are usually pedunculated, having an inner hypodense region surrounded by sclerotic bone. They grow slightly more rapidly than compact osteomas [3,7,18]. Differential diagnosis of osteoma involves benign and malignant lesions: exostoses, osteoid osteoma, benign osteoblastoma, ossifying fibroma, fibrous dysplasia, chondroma,

osteochondroma, calcified meningioma, isolated eosinophilic granuloma, Paget's disease, giant cell tumor, osteosarcoma, and osteoblastic metastasis. Typically, malignant neoplasms show indistinct margins at imaging [2,7].

*4.5. Surgical Intervention*

Surgery is the treatment for mastoid osteoma (Figure 4). As it is a benign lesion, every effort should be made to avoid intra- and postoperative complications. If the tumor is small, with little cosmetic deformity and no symptoms, follow-up with imaging is preferred, with the option of surgical excision. Otherwise, in the presence of pain, neurological alterations, or other complications, surgery is recommended [2,9]. Removal of superficial lesions is a simple procedure with a low risk of complications. It consists of an on-demand postauricular approach to expose the lesion and cleavage of the tumor from unaffected cortical tissue, by a bone chisel, curette, or drilling, depending on the features of the osteoma [1,7,16,17]. There is only one reported case of three-dimensional exoscope-assisted surgery of a giant lesion [19]. Craniotomy and cranioplasty may be considered for greater tumors. Exposure of mastoid air cells during the procedure prompts mastoidectomy with standard closure of the defect [7,16]. In the case of an extended mastoidectomy, reconstruction with titanium mesh can be indicated to prevent cosmetic deformity and maintain an aerated mastoid cavity [2,20]. If osteomas extend into the fallopian canal and bony labyrinth, subtotal excision is preferred to preserve these structures [12]. Osteomas involving the middle and inner ear usually do not grow, and they are treated only if become symptomatic. Likewise, auditory meatus osteomas are partially or totally removed when symptomatic by either middle cranial fossa or a suboccipital approach depending on the features of the lesion [5]. Complications of surgery are facial nerve injury, tearing of the sigmoid sinus, and postoperative discharge [2]. The recurrence rate of mastoid osteomas is very low [2,9], with only two cases of recurrence reported in the literature [13]. Malignant transformation was never reported [7,16].

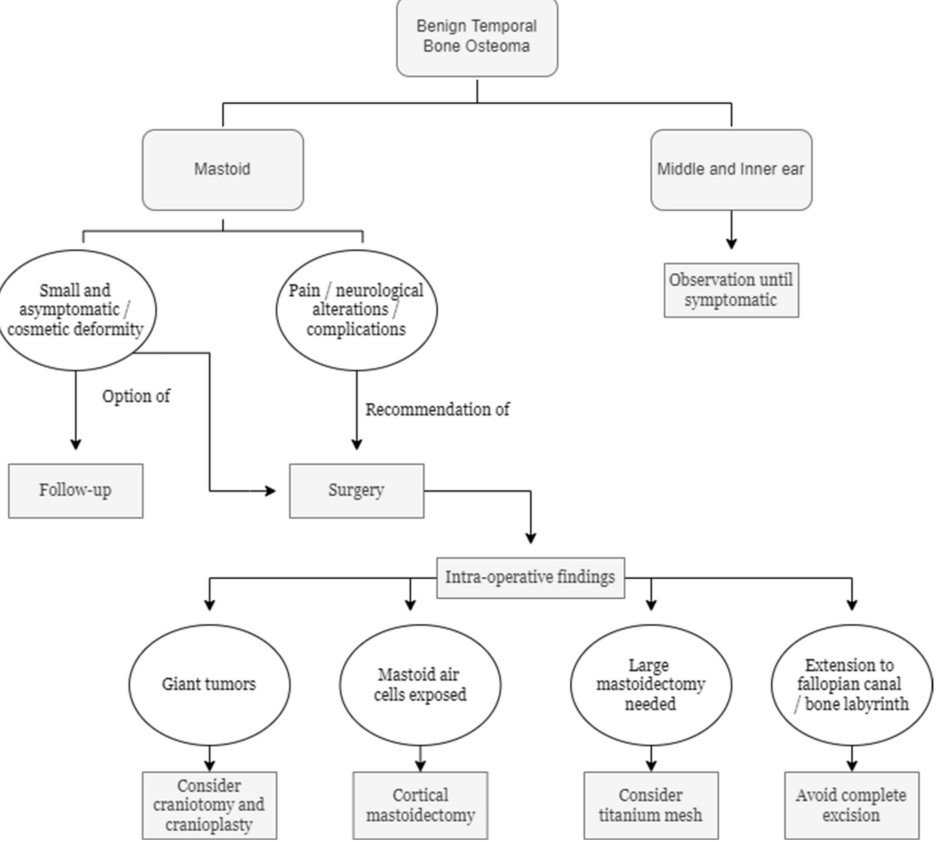

**Figure 4.** Temporal bone osteomas: decisional algorithm.

## 5. Conclusions

Mastoid osteomas are rare, benign, slow-growing, and frequently asymptomatic bony tumors. Other bony lesions of the mastoid region should be considered in the differential diagnosis. Surgery is the treatment of choice and should be performed in the presence of symptoms or for cosmetic reasons.

**Author Contributions:** Conceptualization, L.O.R.d.Z.; validation, G.D. and L.O.R.d.Z.; investigation, G.D. and L.O.R.d.Z.; data curation, G.D. and L.O.R.d.Z.; writing—original draft preparation, G.D.; writing—review and editing, G.D. and L.O.R.d.Z.; supervision, L.O.R.d.Z.; project administration, L.O.R.d.Z. All authors have read and agreed to the published version of the manuscript.

**Funding:** This research received no external funding.

**Institutional Review Board Statement:** Ethical review and approval were waived for this study, due to the collection of observational retrospective data.

**Informed Consent Statement:** Informed consent was obtained from all subjects involved in the study.

**Data Availability Statement:** Not applicable.

**Conflicts of Interest:** The authors declare no conflict of interest.

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
