# Peer review of "Long-Term Surgical Results of Cortical Mastoid Bone Osteomas"

_audiolres, doi:10.3390/audiolres12030030_

Round 1

Reviewer 1 Report

Dear Author, I'm reading the manuscript with interest. It is well done and instructive. Only one suggestion, in the discussion, you can underline that osteoma is a benign disease and the surgery have to avoid injuries. Moreover, a decision tree for the treatment of the osteoma of the temporal bone could be very usefull. 

Author Response

Dear Reviewer, we appreciate and apply in the revised version of the manuscript your suggestions aimed to improve the quality of the manuscript.

Below you will find a detailed explanation of all the changes and additions made in the text, according to your suggestions.

  1. We underlined, in the discussion, that being the osteoma a benign lesion, surgery needs to avoid injuries (lines 110 and 187-188);
  2. a flow chart for treatment decision was added.

Reviewer 2 Report

In this study the authors present three case reports of relatively rare temporal bone osteomas located in the mastoid process. The manuscript is written appropriately, the cases are well-documented and the discussion adequately highlights the importance of the described lesions. However, some minor corrections should be made to improve the quality of the work.

1) In the Introduction, the authors should also refer to reviews or clinical studies on osteomas that support the presented overview (only case reports are cited in the Introduction).

2) Regarding Figures 1 - 3, the osteomas should be clearly marked on the CT scans. Other described morphological features, such as the pedicle in Figure 1, the trabeculae in the deepest component of the osteoma in Figure 2 and the subtle line margin in Figure 3 should also be clearly marked on the images.

3) When first referring to anatomical structures, the authors should add the proper anatomical name (besides the commonly used clinical term) as noted in the Terminologia anatomica in brackets (where relevant); e.g. external auditory canal (EAC, external acoustic meatus). In addition, please clarify the the term "inner table of the temporal bone" - does this refer to the inner lamina or something else?

4) When referring to measurements (primarily on CT scans), if the precision of a measurement is at one decimal point (e.g. 2.4 cm), then all respective measurements should be written to the same degree of precision (e.g. 3.0 cm). 

5) When mentioning an abbreviation the first time (except very common ones, such as CT), the full name should be given with the abbreviation mentioned in brackets (e.g. ABR - auditory brainstem response test?).

6) Minor language corrections are also necessary:

  • lines 39 - 40: the order of adjectives should be the same as in lines 58 - 59 (slowly increasing left-sided postauricular swelling)
  • line 52: "suggestive of" instead of "suggestive for"
  • line 81: "in a few days" instead of "in few days"
  • line 87: "referred due to" instead of "referred for"
  • line 113: "most commonly found" instead of "most found"
  • line 155: "presented" instead of "present"
  • Overall, please check the use of definite and indefinite articles in the text (a/an, the) and consider rewording some sentences to make the meaning clearer (e.g. lines 118 -120). 

Author Response

Dear Reviewer, we appreciate your comments and apply your suggestions as highlighted in the revised version of the manuscript and we sent again the manuscript to our trusted native English proofreader for a detailed check

Below you will find a detailed explanation of all the changes and additions made in the text, according to your suggestions.

  1. in the introduction, a reference to the retrospective study conducted by Abhilasha (2019), was added (line 29);
  2. regarding figures, clear marks have been added as you suggested;
  3. proper anatomical names were added in brackets or changed within the text (lines 26, 28-29, 115, 131); the term "inner table of the temporal bone" (line 132) does refer to the inner lamina and was changed into "internal table of the temporal bone";
  4. CT scan measurement at one decimal point were corrected (line 72);
  5. full names were given before the use of abbreviations (lines 68-69);
  6. minor language corrections were made in lines 40, 55, 85, 91, 118, 168;
  7. sentence in lines 123-124 was reworded to make the meaning clearer.

Reviewer 3 Report

Interesting work.

Author Response

Dear Reviewer, we are very glad of your favorable comment, and we sent again the manuscript to our trusted native English proofreader for a detailed check